# Molecular Imaging of Fibrosis in Benign Diseases: An Overview of the State of the Art

**DOI:** 10.3390/ph17030296

**Published:** 2024-02-26

**Authors:** Yongbai Zhang, Wenpeng Huang, Hao Jiao, Lele Song, Lei Kang

**Affiliations:** Department of Nuclear Medicine, Peking University First Hospital, Beijing 100034, China

**Keywords:** molecular imaging, fibrosis, benign diseases, agent development, fibrosis imaging

## Abstract

Fibrosis is a progressive pathological process participating in the progression of many diseases and can ultimately result in organ malfunction and failure. Around 45% of deaths in the United States are believed to be attributable to fibrotic disorders, and there are no favorable treatment regiments available to meet the need of blocking fibrogenesis, reversing established fibrosis, and curing diseases, especially in the terminal stage. Therefore, early detection and continuous monitoring provide valuable benefits for patients. Among all the advanced techniques developed in recent years for fibrosis evaluation, molecular imaging stands out with its distinct advantage of visualizing biochemical processes and patterns of target localization at the molecular and cellular level. In this review, we summarize the current state of the art in molecular imaging of benign fibrosis diseases. We will first introduce molecular pathways underlying fibrosis processes and potential targets. We will then elaborate on molecular probes that have been developed thus far, expounding on their mechanisms and current states of translational advancement. Finally, we will delineate the extant challenges impeding further progress in this area and the prospective benefits after overcoming these problems.

## 1. Introduction

Fibrosis is a progressive pathological process that is responsive to any types of tissue injury in any organ, which is characterized by the disproportionate accumulation of extracellular matrix (ECM) and can ultimately result in organ malfunction and even organ failure [1,2]. Fibrosis directly or indirectly participates in the progression of many diseases, such as idiopathic pulmonary fibrosis (IPF), chronic obstructive pulmonary disease (COPD), myocardial fibrosis, liver cirrhosis, chronic kidney disease (CKD), Crohn’s disease, systemic sclerosis (SSc), and systemic lupus erythematosus (SLE), and various organs can be involved, including the lungs, liver, kidneys, heart, eyes, and skin (Figure 1) [3,4]. Fibrotic diseases lead to significant morbidity and mortality worldwide. For instance, fibrosis in the eyes can cause the disruption of the highly ordered ocular architecture and lead to disturbed or diminished vision [4]. And it is reported that 45%, of deaths in the United States are believed to be attributable to fibrotic disorders [5]. Moreover, although several antifibrosis drugs have been approved by Food and Drug Administration (FDA) for treatment in different types of fibrotic disorders, including pirfenidone for pulmonary fibrosis therapy, the available regiments can only help relieve symptoms and cannot meet the need for blocking fibrogenesis, reversing established fibrosis, and curing diseases, especially in the terminal stage [6]. Detecting fibrosis in its early stages allows for timely intervention, potentially preventing further progression and preserving organ function. Continuous monitoring can track the progression of fibrosis, assess the treatment efficacy, and make necessary adjustments to therapies. Therefore, early detection and repeatable monitoring provide valuable benefits for patients. Furthermore, considering the heterogeneity of the progression rate, target expression, and target engagement of fibrosis, accurate patient stratification and personalized treatment calls for better tools to detect, phenotype, and quantify crucial biomarkers [7].

In recent years, medical imaging has become increasingly significant in clinical practice for the purposes of diagnosis, staging, and treatment monitoring. This trend is particularly pronounced since the emergence of personalized precision medicine [8]. Imaging technologies offer the advantage of repeated and non-invasive visualization of abnormal tissues. In contrast, obtaining accurate tissue samples through biopsy is challenging due to the spatial and temporal heterogeneity within the fibrosis lesion, making it difficult to capture the complete landscape [9]. IPF can be assessed and diagnosed through high-resolution computed tomography (HRCT) without biopsy in cases with typical radiologic patterns [10]. The late gadolinium-enhanced magnetic resonance imaging (MRI) is usually applied for the detection of myocardial fibrosis in clinical practice [11]. However, these conventional modalities usually have limitations in their ability to solely detect the outcome of fibrosis, providing information on its extent and pattern, but they are unable to characterize the current disease activity or accurately predict the progression of the disease [12].

Molecular imaging is a non-invasive technique that enables the visualization and quantification of biochemical processes and patterns of target localization at the molecular and cellular level in living organisms, which are invisible in conventional anatomical imaging methods. The standardized definition of molecular imaging was first established by the Radiological Society of the Society of Nuclear Medicine and Molecular Imaging (SNMMI) and North America (RSNA) in 2005, defined as techniques “directly or indirectly monitoring and recording the spatiotemporal distribution of molecular or cellular processes for biochemical, biologic, diagnostic, or therapeutic applications” [13]. With a distinct advantage over conventional anatomical imaging methods, molecular imaging stands as the cutting-edge technology in medical imaging and has witnessed remarkable advancements in recent years [14]. It integrates techniques including ultrasound, MRI, CT, nuclear medicine techniques (single-photon emission computed tomography, SPECT, and positron emission tomography, PET), and optical imaging to provide detailed insights into molecular pathways and cellular functions in real time. Molecular probes such as target-specific small molecules, antibodies, and nanoparticles, which are compounds engineered to specifically bind to molecules involved in a biological or pathological pathway, are necessary for molecular imaging [15]. With the refinement of techniques and development of probes, molecular imaging holds great promise in detecting both the outcomes and the disease activity of fibrosis, identifying treatment targets and stratifying patients, thereby enhancing personalized patient care and the development of therapeutic strategies.

Despite its significant advancements, molecular imaging has predominantly focused on oncology. In comparison, research on benign fibrotic diseases remains relatively scarce and insufficient. Additionally, while there have been achievements in the development of techniques and probes for fibrosis, several challenging gaps persist that require attention for effective clinical translation. In this review, we introduce molecular pathways underlying fibrosis processes and potential targets. We then elaborate on molecular probes developed thus far in benign fibrosis diseases, expounding on their mechanisms and current states of translational advancement. Finally, we delineate the extant challenges impeding further progress in this area and the prospective benefits after overcoming these problems.

## 2. Molecular Mechanisms of Fibrosis

Numerous reasons, such as toxins, infections, and drugs can lead to tissue injury, thus initiating and driving the progression of fibrosis. When epithelial and/or endothelial damage occurs, it triggers a series of intricate wound healing processes, facilitating a rapid restoration of homeostasis [3]. The increased endothelial permeability (i.e., vascular leak) and antifibrinolytic coagulation cascade (i.e., extravascular coagulation), which is responsible for the blood clot formation and preventing excessive blood loss, are primarily activated in response to inflammatory mediators that are released by damaged epithelial and/or endothelial cells [16]. Following this, an inflammation and immune activation phase ensues, wherein leukocytes such as macrophages, neutrophils, dendritic cells, and T/B cells are recruited, activated, and induced to proliferate by the chemokines and growth factors (GFs) that are produced by epithelial and/or endothelial cells, platelets, and early inflammatory cells [17,18]. Myofibroblasts and epithelial and/or endothelial cells secrete matrix metalloproteinases (MMPs) that destroy basement membrane, facilitating the recruitment of inflammatory cells to the injury site [18]. The resident or recruited and activated macrophages and neutrophils will eliminate necrotic cellular debris and pathogenic material in the acute stage, and as the tissue repair begins, macrophages promote anti-inflammatory effects, as well as the migration and proliferation of fibroblasts and endothelial cells [19]. Meanwhile, activated leukocytes release profibrotic cytokines and GFs, such as transforming growth factor beta (TGF-β), interleukin 13 (IL-13), and platelet-derived growth factor (PDGF), that promote the recruitment, proliferation, and activation of fibroblasts [18]. 

In pathological fibrosis, fibroblasts serve as the essential source of ECM, and under the influence of mechanical tension and cytokines such as TGF-β, they can differentiate into myofibroblasts expressing α-smooth muscle actin (α-SMA) [17,20]. Myofibroblasts are characterized by their contractile apparatus, heightened responsiveness to chemical signals such as chemokines and GFs, and the capacity to secrete remarkably elevated levels of matrix components, which represent the crucial mediators in all fibrotic diseases [20,21]. Aside from their main contribution, fibroblasts, myofibroblasts may originate from some alternative precursor cells such as resident mesenchymal stem cells, circulating fibroblast-like cells known as fibrocytes that are derived from bone marrow mesenchymal stem cells, and cells undergoing an epithelial/endothelial–mesenchymal transition (EMT/EndMT) process, to compensate for the insufficiency of resident fibroblasts [18,22]. EMT/EndMT is an early and essential event during fibrogenesis. Via this process, specialized epithelial/endothelial cells undergo a phenotypic conversion that leads them to adopt new characteristics as fibroblasts and myofibroblasts. EMT/EndMT provides a crucial source of myofibroblasts and emerges as an important mechanism in the progress of wound healing and organ fibrosis [23]. Radiotracers against known EMT biomarkers like repressed epithelial markers (e.g., E-cadherin, occludins), elevated mesenchymal markers (e.g., vimentin, fibronectin), and other inducers like transcription factors could be explored for detecting EMT using molecular imaging techniques [24]. Shortly after the initial inflammatory phase, myofibroblasts initiate the production of ECM components (collagen type I, fibronectin, elastin) and execute wound contracture. Meanwhile, endothelial cells actively facilitate the formation of new blood vessels [16]. And during the remodeling phase, the provisional deposited ECM is crosslinked and turned over by the action of lysyl oxidase (LOX) and becomes organized [25]. After the blood vessels have been restored, a gradual elimination of scar tissue occurs, creating a conducive environment for epithelial and endothelial cells to undergo division and migration and eventually restore the damaged tissue [25]. However, the presence of persistent inciting factors, such as chronic infection and inflammation, can trigger a prolonged wound healing response, including continuous activation of myofibroblasts and excessive accumulation of ECM components, culminating in the development of fibrosis [16].

The molecular processes driving fibrosis are wide-ranging and complex. Diverse profibrotic and antifibrogenic factors contribute to modulating fibrotic signaling cascades. TGF-β is a major inducer of fibrosis. It drives EMT, activates myofibroblasts, and stimulates ECM production. The TGF-β type Ⅰ/Smad3 pathway holds particular importance, playing a critical role in mediating TGF-β-driven fibrogenesis [1]. Other contributing factors include GFs like PDGF; inflammatory cytokines like TNFα, IL-1β, and IL-6; transcription factors; ECM proteins; integrins; and epigenetic and genetic changes [2]. Together, these factors promote profibrotic signaling, with TGF-β recognized as a major and potent inducer of fibrosis.

In IPF, fibrosis arises as a consequence of aberrant wound healing responses to repetitive alveolar epithelial cell injury, triggering the wound healing cascades that are described above. When appropriately regulated in duration and magnitude, these cascades—involving re-epithelialization, fibroblast apoptosis, and matrix degradation—restore normal lung architecture and function. However, when deregulated or overexuberant, persistent fibroblast activation and matrix deposition culminate in progressive fibrosis and functional decline [20]. Similarly, in CKD, myofibroblasts that are stimulated largely by TGF-β overproduce ECM components, forming scar tissue. This destroys tubules and peritubular capillaries, provokes pro-inflammatory responses, and ultimately impairs kidney function [26]. 

These intricate molecular processes can be roughly divided into four parts: vascular leak and extravascular coagulation, inflammation and immune activation, fibroblast activation and myofibroblast differentiation, and ECM deposition and remodeling (Figure 2). Many biomarkers have been identified and explored for agent development, such as the fibroblast activation protein (FAP) and somatostatin receptor in fibroblast activation and collagen in ECM deposition. Figure 2 highlights some of the most important examples participating in different pathways.

## 3. Molecular Probes for Imaging of Fibrosis and Fibrogenesis

Multiple probes targeting different molecular pathways have been developed and evaluated in preclinical or clinical studies. Table 1 provides a detailed summary of the various probes. 

Diverse imaging modalities are involved, including ultrasound, MRI, CT, nuclear medicine techniques, and optical imaging. Each imaging method is used for different purposes due to its specific imaging ability. Ultrasound provides real-time imaging and cost-effective organ evaluation without radiation exposure but has limited tissue penetration and high operator dependence. MRI offers high soft tissue contrast and spatial resolution, but is limited by long scan times, a high cost, and low contrast agent sensitivity. CT is widely available and featured by its fast acquisition time, high spatial resolution, cost-effectiveness, and relative simplicity. However, extracting molecular information purely from X-ray attenuation differences poses challenges including low sensitivity, limited soft tissue contrast, and high radiation exposure [27]. Compared with other modalities, CT is a premature platform as an emerging molecular imaging technology with room to grow. Recent advances in targeted contrast agents and CT hardware, such as the introduction of novel spectral photon-counting X-ray detectors, do renew the interest in CT’s molecular imaging capabilities [28]. Nuclear medicine imaging like PET and SPECT allow for sensitive functional imaging but require radioactive tracers and lack anatomical detail. Optical imaging provides high contrast agent sensitivity with fluorescent labels but has very limited tissue penetration [29]. A detailed description of the merits and limitations of all molecular imaging modalities exceeds this review, and a review is listed here for further reading [30].

**Table 1 pharmaceuticals-17-00296-t001:** Summary of the probes used in molecular imaging of benign fibrosis diseases.

Probe	References	Molecular Process	Molecular/Cell Target	Stage of Development	Imaging Type	Disease	Potential Clinical Use
Gadofosveset	[31]	Vascular leak	Serum albumin	Human studies (FDA-approved)	MRI	Pulmonary fibrosis	Disease activity
EP-2104R	[32,33]	Extravascular coagulation	Fibrin	Animal studies	MRI	Pulmonary and liver fibrosis	Disease activity, treatment response
[^64^Cu]Cu-DOTA-ECL1i	[34]	Macrophage	CCR2	Animal and human studies	PET	Pulmonary fibrosis	Diagnosis, disease activity, treatment response
BMV109/BMV101	[35]	Macrophage	Cysteine cathepsin	Animal and human studies	PET	Pulmonary fibrosis	Disease activity
[^64^Cu]Cu-LLP2A	[36,37]	Recruitment of immune cells	VLA-4	Animal studies	PET	Pulmonary fibrosis	Disease activity
[^68^Ga]Ga-pentixafor	[38,39]	Recruitment of immune cells	CXCR4	Animal and human studies	PET	Pulmonary and myocardial fibrosis	Disease activity, treatment response, outcome prediction
A20FMDV2	[40,41,42,43]	Activation of TGFβ	Integrin α_v_β_6_	Animal and human studies	PET and SPECT	Pulmonary fibrosis	Disease activity, treatment response
Knottin	[44]	Activation of TGFβ	Integrin α_v_β_6_	Animal and human studies	PET	Pulmonary fibrosis	Disease activity, treatment response
[^68^Ga]Ga-FAPI-04/46	[45,46,47,48,49,50,51,52,53,54,55,56,57,58,59]	Activated fibroblasts	FAP	Animal and human studies	PET	Cardiac diseases, IgG4-RD, renal fibrosis, pulmonary fibrosis, and liver fibrosis	Diagnosis, disease activity, treatment response
[^68^Ga]Ga-MHLL1	[60]	Activated fibroblasts	FAP	Animal studies	PET	Myocardial infarction	Diagnosis, disease activity
[^111^In]In-octreotide scintigraphy	[61,62]	Activated fibroblasts	Somatostatin receptor	FDA-approved	SPECT	Pulmonary fibrosis	Diagnosis
[^68^Ga]Ga-DOTANOC	[63]	Activated fibroblasts	Somatostatin receptor	Human studies	PET	Pulmonary fibrosis	Disease activity
[^99m^Tc]Tc-cRGD	[64]	Activated HSCs	Integrin α_v_β_3_	Animal studies	SPECT	Liver fibrosis	Disease activity
[^99m^Tc]Tc-3PRGD2	[65]	Activated HSCs	Integrin α_v_β_3_	Animal studies	SPECT	Liver fibrosis	Disease activity, treatment response
[^18^F]-Alfatide	[66]	Activated HSCs	Integrin α_v_β_3_	Animal studies	PET	Liver fibrosis	Disease activity
[^18^F]FPP-RGD_2_	[67,68]	Activated HSCs	Integrin α_v_β_3_	Animal studies	PET	Liver and pulmonary fibrosis	Disease activity
RGD-USPIO	[69]	Activated HSCs	Integrin α_v_β_3_	Animal studies	MRI	Liver fibrosis	Disease activity
Den-RGD	[70]	Activated HSCs	Integrin α_v_β_3_	Animal studies	MRI	Liver fibrosis	Disease activity
[^99m^Tc]Tc-CRIP	[71,72]	Myofibroblasts	Integrin α_v_β_3_	Animal studies	SPECT	Myocardial fibrosis	Disease activity, treatment response
EP-3533	[73,74,75,76,77,78,79,80]	Collagen deposition	Type I collagen	Animal studies	MRI	Myocardial, liver and pulmonary fibrosis	Diagnosis, disease activity, treatment response
CM-101	[81,82]	Collagen deposition	Type I collagen	Animal studies	MRI	Liver and post-chemotherapy fibrosis	Disease activity
ProCA32.collagen1	[83]	Collagen deposition	Type I collagen	Animal studies	MRI	Liver fibrosis	Disease activity
SNIO-CBP	[84]	Collagen deposition	Type I collagen	Animal studies	MRI	Liver fibrosis	Diagnosis and disease activity
[^68^Ga]Ga-CBP8	[85]	Collagen deposition	Type I collagen	Animal studies	PET	Pulmonary fibrosis	Disease activity, treatment response
[^64^Cu]Cu-CBP7	[86]	Collagen deposition	Type I collagen	Animal studies	PET	Pulmonary fibrosis	Disease activity
[^99m^Tc]Tc-CBP1495	[87]	Collagen deposition	Type I collagen	Animal studies	SPECT	Pulmonary and liver fibrosis	Disease activity
Collagelin	[88,89,90]	Collagen deposition	Type I and III collagen	Animal studies	SPECT and PET	Myocardial, pulmonary and liver fibrosis animal	Disease activity
PVD	[91]	ECM deposition	Type I collagen	Animal studies	NIRF	Pulmonary fibrosis	Disease activity
CNA35-AuNPs	[92,93]	ECM deposition	Type I collagen	Animal studies	CT	Myocardial fibrosis	Disease activity
CNA35-Cy7	[94]	ECM deposition	Type I collagen	Animal studies	CT–fluorescence imaging	Renal fibrosis	Disease activity
CNA35-PFP NPs	[95].	ECM deposition	Type I collagen	Animal studies	Ultrasound	Myocardial fibrosis	Disease activity
ESMA	[96,97,98,99]	ECM deposition	Elastin	Animal and human studies	MRI	Myocardial, renal and liver fibrosis animal	Disease activity, treatment response
[^89^Zr]Zr-pro-MMP-9 F(ab’)_2_	[21]	ECM deposition	MMPs	Animal studies	PET	Intestinal fibrosis	Disease activity
[^18^F]MAGL-4-11	[100]	ECM deposition	MAGL	Animal studies	PET	Liver fibrosis	Disease activity
Gd-Hyd	[76,101]	Crosslinking	Allysine aldehyde of oxidized collagens	Animal studies	MRI	Pulmonary and liver fibrosis	Diagnosis, disease activity, treatment response
Gd-CHyd	[12]	Crosslinking	Allysine aldehyde of oxidized collagens	Animal studies	MRI	Pulmonary fibrosis	Disease activity
Gd-1,4	[102]	Crosslinking	Allysine aldehyde of oxidized collagens	Animal studies	MRI	Liver fibrosis	Disease activity
Gd-OA	[103]	Crosslinking	Allysine aldehyde of oxidized collagens	Animal studies	MRI	Pulmonary fibrosis	Disease activity
HTCDGd	[104]	Crosslinking	Allysine aldehyde of oxidized collagens	Animal studies	MRI and fluorescence imaging	Liver fibrosis	Disease activity, diagnosis

### 3.1. Vascular Leak and Extravascular Coagulation

Tissue injury causes the disruption of blood vessels, and in conjunction with the impact of platelet degranulation, it induces increased endothelial permeability and extravasation of blood constituents, which are considered some of the early and fundamental responses to tissue injury [25,105]. In the bleomycin mouse model, it has been observed that dysregulated vascular permeability plays a significant role in promoting the advancement of pulmonary fibrosis [106]. Gadofosveset, an FDA-approved gadolinium (Gd)-based contrast agent, reversibly binds to serum albumin and presents prolonged vascular presence and about 5-fold elevated relaxivity (*r*_1_). These properties allow gadofosveset-enhanced MRI to detect increased extravascular albumin concentration and has been demonstrated to effectively identify vascular permeability [107]. Montesi et al. [31] conducted a clinical trial to investigate the potential of gadofosveset-enhanced MRI in detecting vascular leak and identifying the site of active tissue injury in patients with pulmonary fibrosis. Their study revealed that, in comparison to healthy participants, patients with pulmonary fibrosis exhibited heightened albumin extravasation throughout all regions of their lungs, including radiographically normal areas, and indicated gadofosveset-enhanced MRI might illustrate areas that are at risk of developing radiographically apparent fibrosis. This exemplifies the translational application of the albumin-binding probe gadofosveset to evaluate disease progression and guide management in fibrosis patients. However, currently, gadofosveset-enhanced MRI is only approved for vascular imaging in peripheral arterials. Expanding to the clinical application in fibrosis requires generating more safety and efficacy data.

The coagulation cascade, characterized by a sequence of events involving activation of Factor X, thrombin generation, fibrin clot formation, and platelet activation, plays a notable function in not only hemostasis, but subsequent inflammatory and fibroproliferative processes [108]. Over-exaggerated and aberrant coagulation significantly contributes to the progression of fibrosis in various diseases through several pathways including the formation of fibrin clots, which serve as scaffolds for fibroblast migration [109], and the activation of the profibrotic thrombin/PAR1/α_v_β_6_/TGF-β axis, which is triggered by coagulation factors, especially the thrombin and factor Xa [32]. EP-2104R is a Gd-based MR contrast agent that demonstrates remarkable specificity in targeting fibrin, exhibiting an affinity that exceeds 100-fold for fibrin over fibrinogen and 1000-fold for fibrin over serum albumin. In addition, the four GdDOTAGA moieties significantly enhance the molecular relaxivity of EP-2104R [110]. EP-2104R-enhanced MRI is regarded as a promising method for detecting intravascular thrombosis [111,112,113]. Its utility for detection and quantification of extravascular fibrin has been investigated in pulmonary and liver fibrosis [32,33]. In a vascular leak-dependent lung fibrosis mice model induced by low-dose bleomycin in combination with endothelial barrier-disrupting agents, EP-2104R-enhanced ultrashort echo time (UTE) lung MRI was used to visualize and quantify fibrin accumulation in mouse lungs and to establish a correlation between the antifibrotic properties of the thrombin inhibitor dabigatran and the attenuation of fibrin deposition [32]. Additionally, EP-2104R MRI could specifically detect inflammation-associated fibrin in the presence of fibrosis in a liver fibrosis rat model, which indicates that EP-2104R could serve as a biomarker for tissue injury and inflammation and monitor the early progress of liver fibrosis [33].

### 3.2. Inflammation and Immune Activation

The initiation and perpetuation of inflammatory cascades and immune activities actuate and modulate the activation and differentiation of fibroblasts, constituting a pivotal pathological impetus underlying the pathogenesis of fibrosis affecting diverse organs [3]. Inflammatory monocytes and neutrophils are recruited to the damaged tissue by a variety of chemotactic factors that are released by injured epithelial/endothelial cells and platelets, and after the differentiation of these myeloid cells, macrophages and neutrophils eliminate the fibrin clot, invading bacteria, and cellular debris [3]. The navigation of C-C motif chemokine receptor 2-positive (CCR2^+^) inflammatory monocytes from the bone marrow niches towards the damaged sites is guided by a concentration gradient of the C-C motif chemokine ligand 2 (CCL2) [114]. CCR2^+^ monocytes and interstitial macrophages increase in pulmonary fibrosis and produce mediators that promote fibroblast accumulation and ECM deposition, implicating a potential biomarker of inflammation in the profibrotic process [115], and the depletion of circulating monocytes using CCR2-deficient mice has resulted in profound mitigation of pulmonary fibrosis in preclinical models [116]. A Cu-64-labeled peptide-based agent recognized the extracellular loop one (ECL1) of CCR2, enabling non-invasive localization and quantification of the CCR2^+^ inflammatory cell burden and specific monitoring of CCR2 activity associated with fibrosis using PET imaging, as validated in both preclinical animal models and clinical investigations of IPF [34,117]. The advantages of radiotracers based on ECL1i include minimal immunogenic response, easy synthesis, and high stability. In lung tissues that were explanted from pulmonary fibrosis patients, CCR2^+^ cells were observed to aggregate in perifibrotic niches and colocalize with increased radiotracer uptake. In therapeutic models, the antifibrotic agent pirfenidone diminished the infiltration of CCR2^+^ interstitial macrophages, decreased the binding of radiotracers, and attenuated fibroproliferation in mouse lungs, implicating the property of [^64^Cu]Cu-DOTA-ECL1i as a cell-selective marker for identifying subsets of patients with fibrotic diseases who may derive therapeutic benefit from pirfenidone treatment [34]. Likewise, the optical imaging probe BMV109 and the newly designed BMV101 utilizing PET are capable of detecting activated macrophages through selectively binding to the upregulated cysteine cathepsins [35]. The heterodimeric cell adhesion molecule very late antigen-4 (VLA-4) is a critical integrin mediating the cell–cell and cell–matrix adhesions that are required for leukocyte influx and the recruitment of immune cells in inflammatory diseases and the pathogenesis of the inflammation–fibrosis axis [118]. Moreover, elevated levels of vascular cell adhesion molecule-1 (VCAM-1) and fibronectin, both of which function as ligands for the VLA-4, have been reported to participate in the progress of pulmonary fibrotic diseases [119,120]. LLP2A is a peptidomimetic ligand demonstrating a potent binding affinity for the active conformational of VLA-4 coupled, which has proven utility as a targeting agent across several hematological and oncological diseases such as melanoma and myeloma [121,122]. [^64^Cu]Cu-LLP2A PET imaging was performed in both a lipopolysaccharide-induced acute lung injury mouse model [37] and a bleomycin-induced fibrotic lung injury mouse model [36], and [^64^Cu]Cu-LLP2A uptake was significantly and specifically correlated with the expression of multiple inflammatory markers and VLA-4, as well as the ultimate extent of lung fibrosis. [^64^Cu]Cu-LLP2A can be easily synthesized with high purity and specific activity. Its favorable safety profile and dosimetry have been proved in a human study [123]. Chemokine receptor CXCR4 (CXC-motif receptor 4) plays an integral role in cell migration processes [124], and CXCR4-targeted [^68^Ga]Ga-pentixafor is another agent that can detect leukocyte infiltration and recruitment. It demonstrates high affinity and selectivity for CXCR4, along with exceptional in vivo stability and significant and specific concentration at the target sites [39]. In mice with pressure overload-induced heart failure, a [^68^Ga]Ga-pentixafor PET imaging signal was correlated with myocardial inflammation and sequentially increased tissue fibrosis [38]. For IPF patients treated with antifibrotic pirfenidone, the CXCR4 expression on the follow-up PET scan after treatment emerged as an independent prognostic predictor wherein elevated pulmonary CXCR4 signal was associated with markedly worse long-term outcomes [39]. 

Following recruitment and activation, leukocytes elaborate a lot of cytokines and GFs, including TGF-β, which is one of the crucial profibrotic mediators. On the one hand, TGF-β can directly induce the differentiation of fibroblasts into collagen-secreting myofibroblasts [3]. On the other hand, it strongly promotes collagen and fibronectin production and ECM accumulation [125]. Integrin α_v_β_6_ is a cell surface adhesion receptor belonging to the arginine-glycine-aspartic (RGD) integrin subset that is induced on damaged epithelium, and it promotes cell adhesion, triggers intracellular signaling cascades, and activates TGF-β in its local environment [126]. α_v_β_6_ demonstrates highly restricted expression with negligible detection across non-pathological tissues; however, pathological upregulation of α_v_β_6_ has been documented extensively across numerous disease states that are characterized by tissue remodeling, inflammatory responses, and neoplastic progression [127]. A20FMDV2 labeled with [^18^F]FBA ([^18^F]FBA-A20FMDV2) was the first radiotracer that was developed for visualization and quantitation of α_v_β_6_ in vivo [127]. With its favorable affinity and selectivity towards the α_v_β_6_ integrin compared with the other RGD integrins, A20FMDV2 has become one of the most effective and selective α_v_β_6_ ligands to date [128]. In lung fibrosis murine models induced by bleomycin, SPECT/CT using [^111^In]In-A20FMDV2 as the image agent was performed to measure α_v_β_6_ levels [40,41]. Lungs of exposed mice exhibited increased radiotracer uptake compared with control groups, and the radioactivity levels correlated positively with the α_v_β_6_ protein expression. In a human study, [^18^F]FBA-A20FMDV2 was used to compare the expression of α_v_β_6_ between the lungs of pulmonary fibrosis patients and healthy humans [42]. Pulmonary fibrosis patients exhibited a 1.59-fold higher lung volume of distribution (V_T_) and 1.91-fold higher SUVs (standardized uptake values, 95% CI: 1.27, 2.87; *p* = 0.996) of [^18^F]FBA-A20FMDV2 compared to healthy subjects. In a phase 1b, randomized, double-blind clinical trial, [^18^F]FBA-A20FMDV2 was utilized as an imaging biomarker to evaluate the pharmacokinetic profile and confirm the pharmacological target engagement of an integrin antagonist drug candidate, GSK3008348 [43]. The administration of GSK3008348 resulted in a significant reduction in the V_T_ of the radiotracer at 30 min, indicating the utility of [^18^F]FBA-A20FMDV2 as a non-invasive imaging biomarker that is capable of quantifying target engagement and predicting the therapeutic response to α_v_β_6_ inhibition treatment (Figure 3a). Tracers based on knottin, a cystine knot peptide, were radiolabeled and engineered as PET imaging probes for detection of α_v_β_6_, and their utility was evaluated across multiple disease states with elevated α_v_β_6_ [44]. Knottin-based PET tracers exhibit high affinity and specificity for α_v_β_6_. Additionally, the knottin scaffold presents the advantage of highly variable backbone residues, enabling tunable pharmacokinetics and straightforward improvement. High-affinity high-specificity [^18^F]FP-R_0_1-MG-F2 was selected as the optimal clinical translation candidate due to its advantages of high tumor uptake, relatively low uptake in normal tissues, and feasible and reliable radiosynthesis. In IPF patients, the uptake of [^18^F]FP-R_0_1-MG-F2 increased compared with healthy humans and accumulated in abnormal lung tissues.

### 3.3. Fibroblast Activation and Myofibroblast Differentiation

The initiation of the inflammatory cascade and immune response, coupled with the release of profibrogenic mediators, induces the recruitment, proliferation, and activation of tissue-resident fibroblasts and other fibroblast precursor cells. Activated fibroblastic cells synthesize and secrete the fundamental structural constituents of the extracellular matrix, including fibrillar proteins, adhesive proteins, and amorphous proteoglycans, and thus, they play indispensable roles across diverse processes, encompassing extracellular matrix genesis, maintenance, remodeling, cutaneous wound repair, inflammatory responses, neovascularization, and tissue fibrogenesis, occurring in both homeostatic and disease states [129]. In response to biomechanical forces and mediators including TGF-β, fibroblasts can undergo phenotypic activation and differentiation into contractile, α-SMA-expressing myofibroblasts. Myofibroblasts stimulate wound contraction and produce remarkably increased amounts of matrix components, leading to excessive ECM accumulation.

#### 3.3.1. Targeting of Fibroblast Activation Protein

FAP is a type II transmembrane glycoprotein and atypical serine protease of the dipeptidyl peptidase (DPP) family and exhibits a restricted expression pattern that is mainly confined to activated fibroblasts and cancer-associated fibroblasts of neoplastic stroma, whereas its expression is detected at practically undetectable levels in normal adult tissue [130]. In recent years, radiolabeled quinoline-based FAP inhibitors (FAPIs) have been developed as a PET radiotracer targeting FAP-expressing cells, inaugurating a new era in molecular imaging, especially for tumor imaging [131,132]. Given that the tumor stroma is predominantly comprised of cancer-associated fibroblasts (CAFs) and can constitute up to 90% of the total neoplastic mass [133], targeting of FAP for imaging represents a promising strategy for the visualization of epithelial tumors, and a huge amount of studies have validated the favorable performance of FAPIs in cancer detection [134,135,136]. Moreover, when compared to 2-Deoxy-2-[fluorine-18]-fluoro-d-glucose ([^18^F]FDG), the predominant PET radiotracer that has been employed in oncology over the past four decades, FAPIs demonstrate a superior efficacy for tumor localization, cancer diagnosis, discrimination of malignant from benign masses, and determination of accurate tumor staging [137,138]. 

As for non-malignant diseases, notwithstanding the relatively limited number, imaging using FAPIs as radiotracers for assessing mesenchymal stromal activation is witnessing rapid progress. Owing to its advantageous characteristics, including low nanomolar affinity, near-complete internalization exceeding 90%, and rapid clearance from circulation, [^68^Ga]Ga-FAPI-04 represents the most prevalent and promising option within FAPI agents [45]. The utility of FAPI PET in non-tumor diseases was first evaluated in a rat model of MI (Figure 4A) [46]. In MI rats induced by coronary ligation, intense tracer accumulation was observed within the MI territory, localized predominately in the border zone of the infarcted myocardium, which was found harboring 3- and 8-fold higher FAP-positive fibroblast densities compared to the infarct center and remote area, respectively [46]. In a large retrospective human study including 229 patients of two cohorts, a multivariate analysis demonstrated that increased [^68^Ga]Ga-FAPI-04 signal intensity was associated with cardiovascular risk factors, diabetes mellitus type II history, some chemotherapy exposure, and a history of radiation to the chest, although without further validation of fibroblast activation or myocardial fibrosis [47]. In a prospective clinical trial with 35 patients, cardiac MR (CMR) and [^68^Ga]Ga-FAPI-46 PET/CT scans were conducted following acute MI, and findings revealed that the area with an elevated tracer uptake extends beyond the infarct region, and an early FAP signal was correlated with a subsequent impairment of left ventricular ejection fraction, intimating that it might be a biomarker of left ventricular remodeling [48]. Additionally, [^68^Ga]Ga-FAPI PET/CT can visualize fibrotic remodeling of the right ventricle in patients with pulmonary arterial hypertension with a correlation between the elevated signal and a dysfunctional right ventricle [49].

FAPI PET/CT was also used to assess activated fibroblasts in IgG4-RD [50], renal fibrosis [51,52,53], pulmonary fibrosis [54,55], liver fibrosis [56], Crohn’s disease [57], and systemic sclerosis (SSc) (Figure 4B,C) [58,59]. FAPI PET/CT shows promise in furthering disease diagnosis, monitoring disease activity and progression, and evaluating treatment response. In 13 patients with IgG4-RD, FAPI PET/CT detected 18 involved organs such as the pancreas and salivary gland, which showed negative tracer uptake in [^18^F]FDG PET/CT, and these findings indicated the superiority of FAPI PET in depicting disease involvement [50]. In a prospective clinical trial conducted by Röhrich et al. [54], 15 patients with fibrotic interstitial lung diseases (fILD) and suspected lung cancer (LC) received [^68^Ga]Ga-FAPI-46 PET/CT scans. A markedly elevated uptake was observed in both fILD and LC lesions, with SUVmax of 11.12 ± 6.71 and 16.69 ± 9.35, respectively. And in a patient diagnosed with a progressive phenotype, the intensively positive tracer accumulation might correlate with the observed clinical progression, suggesting the utility of FAPI PET in differentiating activated, progressive fibrosis from inactive one. In a human-sized swine model with liver fibrosis, [^68^Ga]Ga-FAPI PET/MRI was used to evaluate the stage of liver fibrosis. Their results displayed liver [^68^Ga]Ga-FAPI uptake that was strongly correlated with the METAVIR score and collagen-proportionate area, calculated based on historical analysis, indicating that [^68^Ga]Ga-FAPI PET can play an important role in staging liver fibrosis [56]. In another prospective study by Zhou et al., [^68^Ga]Ga-FAPI-04 PET/CT was implemented in a cohort of patients with renal fibrosis to ascertain the disease extent. Statistically significant divergences in maximum SUVmax and target-to-background ratios (TBR) were revealed between subjects with mild versus severe fibrotic burden [51]. Langer et al. engineered a novel FAP-targeted radiotracer [^68^Ga]Ga-MHLL1 with a simplified one-step-synthesis accessible precursor and efficient labeling. In both mouse and human tissues, it displayed specific binding to FAP-positive cells. In an in vivo study using an MI mouse model, significant elevated uptake was observed in infarcted myocardium regions compared with non-infarcted regions, demonstrating its desirable efficacy for imaging fibroblast activation in MI mice [60].

#### 3.3.2. Targeting of Somatostatin Receptor

The somatostatin receptor (SSTR) is another target for detecting activated fibroblasts. The baseline expression of SSTRs is relatively low in normal tissue but exhibits robust upregulation in epithelial, endothelial, and inflammatory cells and fibroblasts within fibrotic lesions. The antifibrotic activity of somatostatin and its analogs is conferred via binding to SSTRs [139]. Octreotide, a synthetically derived somatostatin analogue, exhibits potent and selective binding affinity for somatostatin SSTR, especially receptor subtypes 2 (sst2) [140]. [^111^In]In-octreotide scintigraphy, currently approved by the FDA, can detect elevated sst2 in patients with IPF, pulmonary fibrosis associated with SSc and sarcoidosis [61,62]. However, the use of [^111^In]In-octreotide scintigraphy is mainly focused on the diagnosis of neuroendocrine malignancies. The evidence supporting the utility of this technique in fibrotic diseases remains insufficient, mainly limited to small exploratory studies, and there is still a gap in terms of clinical application. [^68^Ga]Ga-DOTA peptides targeting SSTRs have emerged as PET tracers, with [^68^Ga]Ga-DOTA-NaI-octreotide (DOTANOC) being distinguished by its broad binding affinity across SSTR subtypes and favorable dosimetric profile compared to alternate tracers [141]. In a study including IPF and nonspecific interstitial pneumonia subjects, a correlative relationship between the radiotracer uptake and disease extent was only observed among the idiopathic pulmonary fibrosis subjects [63]. SSTR PET/CT is also proposed as capable of distinguishing between the acute and chronic phase in cardiac sarcoidosis, since in the chronic fibrotic stage, fibrosis is formed which is deficient in SSTR expression compared to the acute inflammatory state [142]. Patients with cardiac sarcoidosis underwent [^68^Ga]Ga-DOTANOC cardiac PET/CT and CMR. While the characteristic pattern of late gadolinium enhancement (LGE) on CMR could not specifically differentiate between inflammation and fibrosis, [^68^Ga]Ga-DOTANOC PET/CT might be better at identifying patients with active inflammation, indicating a complementary role to CMR [142].

#### 3.3.3. Targeting of Integrin α_v_β_3_

Hepatic fibrogenesis is characterized by the differentiation of quiescent vitamin A-storing hepatic stellate cells (HSCs) into activated myofibroblast-like cells in response to chronic liver injury, resulting in the unrestrained proliferation of these fibroblasts and the overproduction of ECM components, leading to the distortion of hepatic architecture [143]. Integrin α_v_β_3_ is expressed by HSCs during their activation and binding of this integrin to extracellular matrix ligands triggers signaling cascades that enhance HSCs’ resistance to apoptosis and increase HSC proliferation [144]. The α_v_β_3_ integrin binds to ECM proteins via the tripeptide sequence of RGD. Radiotracers based on cyclic RGD peptides (cRGD) have been developed for targeting α_v_β_3_. In rat models induced by thioacetamide (TAA) or carbon tetrachloride (CCl_4_), [^99m^Tc]Tc-labeled cRGD was proposed as a SPECT tracer for non-invasive molecular imaging of α_v_β_3_ expression to detect HSC activity and was demonstrated to be able to specifically bind activated HSC with high affinity and abundant receptors [64]. Moreover, quantitative analysis revealed a significant positive correlation between the tracer signal and the severity of liver fibrosis [64]. Another SPECT RGD peptide-based radiotracer, [^99m^Tc]Tc-3PRGD2, which exhibited high affinity and easy access, was used in a TAA-induced liver fibrosis rat model to monitor the progression and recovery of liver fibrosis. In both the group of rats with spontaneous recovery and the group receiving antifibrotic treatment with IFN-α, a significant decrease in the liver-to-background ratio of radioactivity was observed [65]. [^18^F]-Alfatide is a PET tracer based on dimeric RGD peptide exhibiting a relatively simple radiosynthesis profile and improved specific binding to the integrin α_v_β_3_ receptor utilizing multivalent affinity enhancement [66]. Murine models with induced hepatic fibrosis through the administration of CCl4 or bile duct ligation (BDL) displayed significantly elevated mRNA and protein levels of integrin α_v_β_3_ and its signaling complex and increased radiotracer retention in fibrotic hepatic tissue following intravenous injection of [^18^F]-Alfatide compared to control mice without liver injury, as assessed by PET [66]. Rokugawa et al. used another dimeric RGD-based PET tracer, [^18^F]FPP-RGD_2_, for detecting integrin α_v_β_3_ expression to evaluate the disease progression in a non-alcoholic steatohepatitis (NASH) mouse model [67]. [^18^F]FPP-RGD exhibited improved target affinity, a relatively low background signal, and rapid clearance. Mice were fed a choline-deficient, L-amino acid-defined, high-fat diet (CDAHFD) for 3 or 8 weeks, and although fibrosis was observed only in mice fed for 8 weeks, SUV was increased in both the 3- and 8-week-feeding groups compared with the respective controls. And the elevated uptake correlated well with the mRNA and protein levels of integrin α_v_ and β_3_. Ultrasmall superparamagnetic iron oxide nanoparticles (USPIO) modified by RGD (RGD-USPIO) were designed as a molecular MR T2 contrast agent and were used for staging liver fibrosis in rat models induced by CCl_4_ [69]. A significant difference was observed in the T2* relaxation rate change (ΔR2*) among rat treatment cohorts receiving CCl_4_ for 0, 3, 6, and 9 weeks. Dendrimer nanoprobes labeled with pentapeptide cRGDyK (Den-RGD) is another MRI tracer that was developed with high specificity and favorable safety profiles. Similar results were observed in TAA-induced liver fibrosis mice [70]. 

RGD-based radiotracers have been developed for other fibrotic diseases, including IPF and myocardial remodeling after MI, by targeting myofibroblasts [68,71,72]. PET/CT utilizing [^18^F]FPP-RGD_2_ as a radiotracer was also implemented in a lung fibrosis rat model to assess IPF, and it turned out that the V_T_ values for [^18^F]FPP-RGD_2_ were strongly correlated with the histopathological and immunohistochemical markers of fibrosis severity, α_v_ expression, and oxygen partial pressure [68]. Using a [^99m^Tc]Tc-labeled Cy5.5-RGD imaging peptide (CRIP), researchers evaluated post-infarction myocardial interstitial remodeling via imaging the α_v_β_3_ on myofibroblasts and found that the maximum CRIP exhibited peak uptake within the infarct area, and antiangiotensin and/or antimineralocorticoid intervention could decrease the CRIP uptake (Figure 3b), indicating that the targeting of myofibroblast α_v_β_3_ integrin expression enables non-invasive assessment of the efficacy of antifibrotic therapies [71,72].

### 3.4. ECM Deposition and Remodeling

Fibrosis is characterized by the pathogenic accumulation of fibrillar collagen and other structural ECM components including fibronectin, elastin, and proteoglycans [2]. Succeeding the deposition of provisional ECM, the remodeling phase ensues, and deposited ECM starts to crosslink and turn over, contributing to increased matrix stiffness and driving fibrosis progression [20].

#### 3.4.1. Targeting of Collagen

The molecular imaging agent of MRI requires robust sensitivity. The high levels of collagen, especially type I collagen, within established fibrotic lesions facilitate sufficient contrast generation to enable sensitive visualization of fibrotic tissues using appropriately designed collagen-targeted MRI contrast agents [73]. EP-3533, a gadolinium-based contrast agent, was derived from a collagen-specific cyclic peptide that was identified through phage display and engineered to have increased binding affinity and improved pharmacokinetic profiles. The utility of EP-3533 was first evaluated in an MI mouse model [73]. The post-injection MRI scans exhibited hyperintense signaling specifically within the infarct territory. The spatial extent and distribution of myocardial fibrosis in histology closely matched the zones of hyperintense signaling on EP-3533-enhanced MR images [74]. EP-3533 has been also used for detecting liver fibrosis, and studies have shown that it can identify fibrosis tissues [75], especially the early onset of fibrosis [76,77], determine the stage of liver fibrosis (Figure 5A) [78], detect fibrosis heterogeneity [79], and monitor the treatment response [80]. However, EP-3533 based on the linear gadolinium chelate Gd-DTPA is unsuitable for clinical use due to the risk of nephrogenic systemic fibrosis from gadolinium retention [81]. Therefore, CM-101, which uses a more stable macrocyclic gadolinium chelate, was developed for targeting collagen [81]. In rodent liver fibrosis models, CM-101 demonstrated rapid blood clearance, low retention of gadolinium in tissue, negligible accumulation in bone, and favorable detection of fibrosis. And in a pancreatic ductal adenocarcinoma murine model with post-chemotherapy fibrosis, the CM-101 MR signal retention increased in the chemotherapy-treated tumors with more severe fibrosis compared with untreated controls [82]. A protein MRI contrast agent, ProCA32.collagen1, has also been developed to address the safety risk of metal toxicity [83]. This agent was engineered by conjugating a type I collagen-targeting peptide to the C-terminus of the protein-based contrast agent ProCA32. It exhibited reduced administered dose requirements, robust resistance to transmetallation reactions, and excellent metal selectivity for Gd^3+^, and it concurrently demonstrated high dual relativity for both *r*_1_ and *r*_2_ and a high binding affinity to collagen type I in animal models of liver fibrosis and NASH. Researchers have also engineered some gadolinium-free molecular MRI probes [84,145]. The performance of a single-nanometer iron oxide nanoparticle-based agent conjugated with collagen-binding peptide SNIO-CBP was evaluated in CCl_4_-induced liver injury or CDAHFD NASH mouse models [84]. A 2.5-fold reduced dose of SNIO-CBP relative to CM-101 achieved commensurate diagnostic performance in rapid detection of liver fibrosis using T1-weighted MRI.

Moreover, some molecular imaging probes have been developed for PET and SPECT to facilitate non-invasive visualization of collagen. A PET probe, [^68^Ga]Ga-CBP8, was developed by modifying a known collagen-specific peptide and was demonstrated to have high specific binding, strong target-to-background contrast, and favorable dosimetry and safety profiles in a bleomycin- and vascular leak-induced mouse pulmonary fibrosis model, namely, a low-dose bleomycin vascular leak (LDBVL) model [85]. [^68^Ga]Ga-CBP8 PET was also used for monitoring the antifibrotic effects of the α_v_β_6_ antibody, 3G9, and mice treated with 3G9 showed reduced uptake and alleviated fibrosis. In another study conducted by Désogère et al., collagen-binding peptide-based [^64^Cu]Cu-CBP7 optimized five other copper-chelated PET probes and showed the highest uptake in fibrotic lungs, as well as superior TBR and enhanced metabolic stability (Figure 5B) [86]. CPKESCNLFVLKD is a fragment sequence of the precursor of MMP-2 which interacts with type I collagen and is identified as an original collagen-binding peptide (designated as CBP1495). Radiolabeled with [^99m^Tc]Tc, it was used for SPECT imaging in lung or liver fibrosis rat models and exhibited high affinity for collagen and low dosimetry [87]. Glycoprotein VI (GPVI), an immunoadhesin collagen receptor, exhibits robust affinity and high specificity for type-I and type-III collagens. A GPVI-mimic peptide, designated collagelin, and its analogs were designed and radiolabeled for targeting collagen using SPECT or PET in myocardial, pulmonary, and liver fibrosis animal models [88,89,90]. 

Platelets can adhere to collagen directly through collagen-binding receptors, such as GPVI and integrin α_2_β_1_ [146]. Platelet-derived nanovesicles labeled with dye (PVD) represent a biomimetic platelet platform that is designed to evade immune recognition and retain native binding properties by preserving the platelet adhesion molecules. It was developed for detecting pulmonary fibrosis utilizing near-infrared fluorescence (NIRF) imaging [91]. In an IPF and hypertension rat model induced by monocrotaline (MCT), PVD was capable of specifically binding collagen and directly detect IPF in the early stage.

CNA35 is a collagen adhesin and has excellent affinity for collagen I [147]. With an approximately 5-fold smaller size compared to antibodies, CNA35 demonstrates enhanced tissue penetration and improved binding kinetics. Gold nanoparticles (AuNPs) functionalized with CNA35 are employed as a CT contrast agent targeting collagen within myocardial scars [92,93]. Specific signal enhancement was detected in the myocardium scar in rats compared with control groups and was associated with histological findings. Labeled with near-infrared fluorophore Cy7, CNA35 was employed as an imaging agent for hybrid computed tomography–fluorescence molecular tomography in animal renal fibrosis models (Figure 5C) [94]. Elevated accumulation of CNA35-Cy7 was observed in fibrotic kidneys in optical imaging. In a subsequent ex vivo study, immunofluorescence analysis revealed the colocalization of CNA35-FITC (fluorescein isothiocyanate) with collagen type I and III deposition in fibrotic renal tissue, and quantitative analysis indicated a significantly augmented expression of collagen in the perivascular areas during the progression of renal fibrosis. Zhou et al. designed a CNA35-based ultrasound molecular probe to image myocardial fibrosis in rabbits [95]. Once attached to myocardial type I collagen, CNA35-labeled perfluoropentane nanoparticles (CNA35-PFP NPs) underwent liquid-to-gas phase transition when subjected to low-intensity focused ultrasound (LIFU) irradiation, which significantly enhanced the ultrasound contrast in the fibrotic area. These phase-changeable nanoparticles were demonstrated to efficiently traverse the vascular endothelium and selectively accumulate in myocardial fibrous tissue.

#### 3.4.2. Other Targets Associated with ECM Deposition

Besides collagen, elastin has been identified as a crucial ECM protein which is upregulated in advanced fibrosis [148]. Gadolinium-based ESMA is developed as a promising probe for specific elastin protein imaging with features of high agent contrast, sensitivity, and safety. It has been shown to facilitate non-invasive assessment of MI, liver fibrosis, and kidney fibrosis [96,97,98,99]. Murine livers subjected to CCl_4_ exhibited significant contrast enhancement upon administration of ESMA, with distinct perivascular signals detected within large and medium-sized vessels [98]. Sun et al. evaluated the capability of Gd-ESMA for detecting elastin deposition in multiple mouse models of renal fibrosis and in fibrotic human kidneys [99]. They confirmed the upregulation of elastin expression in ten rat or mouse models and fibrotic human kidneys ex vivo. Then they proceeded to implement ESMA MRI in three mouse models of renal fibrosis, which were adenine nephropathy, unilateral ureteral obstruction, and ischemia/reperfusion injury, and revealed elevated ESMA accumulation in diseased kidneys. Additionally, they identified the efficacy of ESMA MRI for longitudinal assessment of disease advancement and therapeutic efficacy. In two mouse models treated with inflammasome inhibitor CRID3 or receptor tyrosine kinase inhibitor imatinib, the obtained MRI signal intensities demonstrated statistically significant reductions compared to control mice. These observations underscore the potential diagnostic and monitoring applications of ESMA-enhanced MRI for non-invasive assessment of disease progression and treatment responses in renal fibrosis.

Various radiolabeled probes have been engineered to selectively bind mediators that are implicated in regulating ECM deposition. MMPs constitute the predominant enzymes mediating the extracellular degradation of ECM. However, divergent from their matrix-degrading activities, some MMPs may potentiate profibrotic processes through pathways that are uncoupled from ECM proteolysis [149]. [^89^Zr]Zr-labelled F(ab’)_2_ antibody fragments targeting pro-MMP-9 ([^89^Zr]Zr-pro-MMP-9 F(ab’)_2_) were used for detecting intestinal fibrosis induced by colitis in a mouse model [21]. The antibody fragments were derived from intact pro-MMP-9-specific antibodies via cleavage of the FC effector region. This process retained the target specificity of the antibody while decreasing the molecular weight, thereby facilitating tissue penetration and enhancing excretion. In contrast to the substantial diminution in expression levels of other MMPs from the inflammatory to the fibrotic phase, they persisted at comparable concentrations in fibrotic and inflamed colonic tissue. It was also demonstrated that the strong signal exists in the kidney and is correlated with colon fibrosis, which was revealed to be renal fibrosis, indicating the capability of [^89^Zr]Zr-pro-MMP-9 F(ab’)_2_ PET in non-invasive detection of disease in organs that are distant from the primary disease localization. Monoacylglycerol lipase (MAGL) represents an inflammatory enzymatic mediator in the degradation of the endogenous cannabinoid receptor ligand 2-arachidonoylglycerol and takes part in inducing HSC activation and ECM accumulation during chronic liver injury [100,150]. In liver fibrosis mouse models induced by BDL or CCl_4_, PET using a probe targeting MAGL, the F-18-labeled MAGL inhibitor ([^18^F]MAGL-4-11), was performed to evaluate the expression levels of MAGL, and significant signal diminution was observed at the early stage of liver fibrosis, which further decreased with disease progression [100]. And histological analysis confirmed this longitudinal decline in both animal and human liver tissues.

#### 3.4.3. Targeting of Oxidized Collagens

Lysyl oxidase (LOX) and lysyl oxidase-like proteins (LOXLs) are critical enzymes of the initial covalent crosslinking of collagens and elastin, catalyzing the oxidization of the collagen lysine residues to allysine aldehyde, which serves as a biomarker of the active fibrogenesis [151]. Hydrazide undergoes a condensation reaction with aldehydes, conferring the capacity for specific binding with oxidized collagens. A hydrazine-equipped aldehyde-targeting MRI contrast probe, Gd-Hyd, was designed and evaluated in pulmonary and liver fibrosis mouse models [101]. Researchers identified the capability of Gd-Hyd-enhanced MRI for fibrosis diagnosis and disease progression and treatment response monitoring in both fibrotic models. It could not only track disease progression and resolution, but also differentiate dynamic fibrogenic remodeling from established fibrotic scars, with significantly decreased intensities at the point of 4 weeks after bleomycin injury, when fibrogenesis ceased, but the fibrotic scar remained. In a study comparing four advanced MRI techniques, including molecular MRI with probe EP-3533 and Gd-Hyd, MR elastography, and native T1 in a rat NASH model, Gd-Hyd MRI scored the highest accuracy in identifying responders and nonresponders in the treated groups [76]. Gd-CHyd is an improved probe that is derived from Gd-Hyd by substituting the reactive hydrazine moiety from a hydrazide to alkyl hydrazine, which allows for greater reactivity and affinity for aldehydes [12]. In bleomycin-induced pulmonary injury mice, significantly higher lung-to-liver contrast and slower lung clearance were observed compared with Gd-Hyd. Another improved MR probe, Gd-1,4, was designed with two hydrazine moieties, and its dual binding with allysine aldehyde resulted in a faster on-rate, slower off-rate, and higher protein-bound relaxivity compared with monobinders [102]. Compared with Gd-CHyd and another dual hydrazine-equipped probe, Gd-1,7, Gd-1,4 exhibited slower liver clearance rates, a favorable safety profile, and the highest change in liver-to-muscle and contrast-to-noise ratio. High concentrations in fibrotic livers were observed in several murine models of liver fibrosis, NASH, and human liver tissues, which were negative in healthy livers. An oxyamine reacts with an aldehyde and forms oxime, which displays greater stability to hydrolysis than the analogous hydrazone or imine. Through the substitution of an oxyamine functional moiety in lieu of the hydrazide constituent in Gd-Hyd, the novel agent Gd-OA was engineered, and it demonstrated rapid uptake, rapid background clearance, and high specificity in a bleomycin-injured mouse model [103]. A hyaluronic acid-derived agent conjugated with oxyamine, Gd-DOTA, and fluorescence Cy5.5 (HTCDGd) was designed for MR and fluorescence imaging of liver fibrosis [104]. It exhibited high specificity and affinity to allysine aldehyde and a capability for early diagnosis and accurate staging of liver fibrosis.

## 4. Current Challenges and Future Directions

In recent years, the field of molecular imaging of benign fibrotic diseases has seen rapid advances, with the emergence of diverse probes targeting many pathogenic pathways. However, despite this considerable interest and endeavor, only a few agents have bridged the gap between theoretical potential and clinical translation to achieve implementation in patient care. 

The development of new probes remains an important area of research to improve molecular imaging capabilities. In order to develop probes identifying targets of interest, researchers need to employ knowledge of target structure and binding sites to rationally design probes that will specifically bind the target. High-throughput screening of compound libraries facilitates rapid testing of thousands of agents and the identification of probes with high affinity. New technology such as antibody engineering, aptamer technology, and nanoparticle platforms provide powerful tools for the development of probes with a high target affinity. Additionally, existing probes can be modified by adding reporter groups like fluorophores or radionuclides for optimizing their binding affinity and imaging properties.

The rigorous validation of potential novel targets and probes for molecular imaging necessitates a comprehensive interrogation encompassing their sensitivity, specificity, in vivo pharmacokinetic characterization, safety profiling, and efficacy determination. Only a few studies that have been undertaken to date, whether preclinical or clinical, have dug into these issues with the requisite degree of rigor and depth. Considerations such as the small sample size, limited clinical study number, and the difference between pathological animal models and actual human diseases exist, and they can jeopardize the methodological rigor and applicability of encouraging outcomes that are obtained from these conducted investigations. Different animal models can engender different conclusions. The observed discrepancy between diverse animal models might also approximate the heterogeneity that is anticipated in human subjects and further underscores the necessity for more standardized and rigorous validation among various models [77]. Moreover, certain pathological progressions cannot be replicated in animal models. For example, in an animal study comparing the efficacy of EP-3533- or Gd-Hyd-enhanced MRI, MR elastography, and native T1, the rats received CDAHFD to establish a NASH disease model [76]. However, this CDAHFD model was deficient in reproducing the metabolic derangements typifying patients with NASH, and it had proven hard to simultaneously establish metabolic dysregulation and steatohepatitis with current techniques [76]. Thus, the undertaking of clinical investigations among large groups of patients is an urgent and indispensable imperative for the further development of these molecular probes. In addition to the efficacy of engineered probes that are validated in proof-of-concept studies, their potential clinical superiority over established imaging modalities remains to be substantiated, including improvements in diagnostic performance, reductions in the economic burden, decrease in risk, and/or enhanced patient comfort.

The overriding regulatory environment represents one of the critical rate-limiting factors engendering delayed translation of molecular imaging techniques from bench to bedside. In the US, despite sub-pharmacologic mass doses, the radiotracers face regulatory constraints that are analogous to those imposed on therapeutic agents [14]. Streamlining regulatory processes will enable more expeditious undertaking of phase I/II trials and clinical translation of these molecular imaging probes that are currently on the bench. 

The development of artificial intelligence (AI) and high-throughput big data analytics portends immense potential to propel molecular imaging into an unprecedented era of progression. By leveraging techniques including deep learning algorithms, the automation of image reconstruction, processing, and analysis can accelerate novel biomarker discovery and probe the validation of fibrosis imaging. However, rigorously validating the robustness and reproducibility of AI-based imaging techniques remains an imperative prerequisite. Expansive datasets will be requisite to empower the training of AI algorithms.

Recently, spatial molecular imaging has emerged as a set of techniques that allow for the visualization and characterization of tissue’s molecular architecture at a subcellular resolution in tissue samples and living subjects. It has been demonstrated to allow for sensitive and specific mapping of biomolecules while minimizing background signals [152]. Since the current research on spatial molecular imaging is still limited to sample tissues and has not yet been applied to living subjects, we do not incorporate it in this review. But we believe continued advancement of spatial molecular imaging methodologies and their integration with conventional modalities promise to provide unique insights complementing traditional molecular imaging. 

## 5. Conclusions

With rapid development these years, molecular imaging has shown immense potential to transform the detection and characterization of fibrotic diseases. A lot of innovative targeting probes have emerged to visualize important pathways underlying fibrosis progression, such as inflammation, fibroblast activation, and ECM remodeling. Nevertheless, rigorous validation of probe performance in clinical populations remains imperative to fully realize the potential of these agents for improving diagnosis, patient stratification, prognostication, and therapeutic efficacy monitoring in fibrosis. Although challenges persist, molecular imaging is poised to fill a critical unmet need in the clinical management of fibrosis.

## Figures and Tables

**Figure 1 pharmaceuticals-17-00296-f001:**
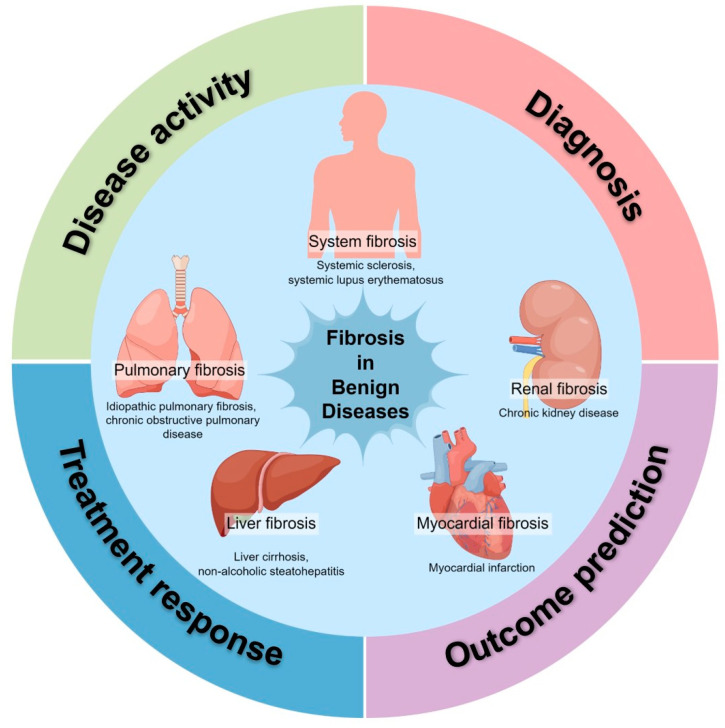
The application of molecular imaging in benign fibrosis diseases. Fibrosis is a progressive pathological process responsive to any type of tissue injury in any organ, such as kidney, lung, heart, liver, and system. It directly or indirectly participates in the progression of many diseases. Molecular imaging can be utilized in diagnosing fibrosis, assessing disease activity and treatment response, and predicting outcome in fibrotic diseases. Illustrated by Figdraw.

**Figure 2 pharmaceuticals-17-00296-f002:**
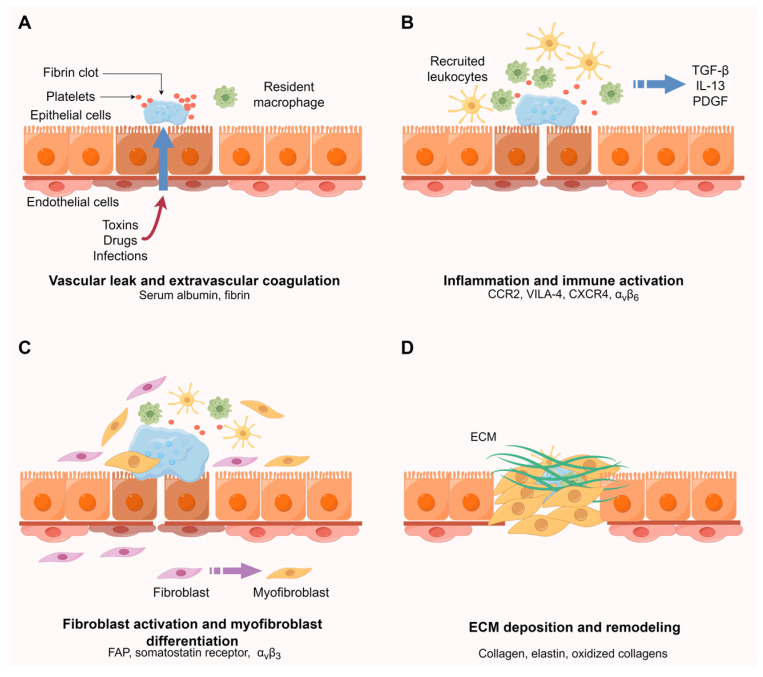
Presentation of molecular mechanisms of fibrosis and important targets. (**A**) Epithelial and/or endothelial injury induced by numerous reasons leads to vascular leak and extravascular coagulation which is responsible for the blood clot formation. (**B**) Leukocytes such as macrophages, neutrophils, dendritic cells, and T/B cells are recruited, activated, and induced to proliferate by the chemokines and growth factors (GFs). (**C**) Fibroblasts are activated and differentiate into myofibroblasts. (**D**) Myofibroblasts initiate the production of ECM components and execute wound contracture. The provisional deposited ECM is crosslinked and turned over by the action of lysyl oxidase (LOX) and becomes organized. Illustrated by Figdraw.

**Figure 3 pharmaceuticals-17-00296-f003:**
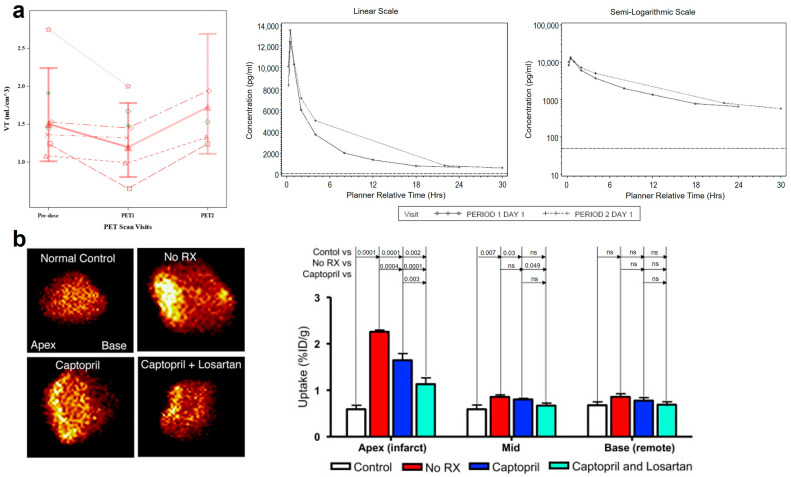
Imaging of targeting at FAP in different diseases. (**a**) Summary of adjusted medians of uncorrected VT, (mL/cm^3^) of [^18^F]FBA20FMDV2 in IPF patients, and median plasma GSK3008348 concentration–time plot. PET1: PET scan on day 1 at ~30 min post-dose; PET2: PET scan on day 2 at ~24 h post-dose [43]. (**b**) Imaging of [^99m^Tc]Tc-CRIP SPECT in a myocardial infarction model. Captopril alone and with losartan significantly reduced tracer uptake [71].

**Figure 4 pharmaceuticals-17-00296-f004:**
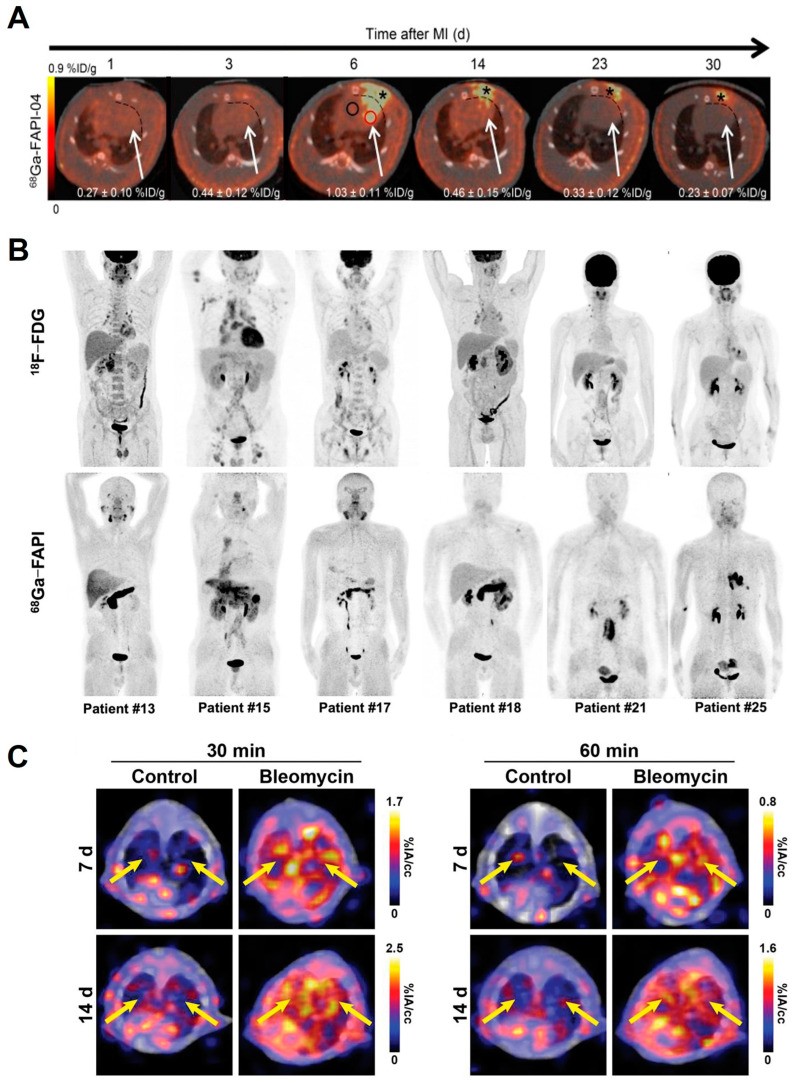
Imaging of targeting of FAP in different diseases. (**A**) Imaging of [^68^Ga]Ga-FAPI-04 uptake for longitudinal monitoring in an MI rat model [46]. Representative regions of interest (2-dimensional) drawn over infarct border zone and remote myocardium are illustrated as red and black circles, respectively. [^68^Ga]Ga-FAPI-04 exhibited elevated uptake in scars from operation (asterisk). (**B**) Imaging of [^68^Ga]Ga-FAPI-04 and [^18^F]FDG uptake in 6 patients with IgG4-RD [50]. [^68^Ga]Ga-FAPI-04 showed superiority to [^18^F]FDG in depicting involvement of the pancreas, bile duct/liver, and salivary gland. (**C**) Imaging of [^68^Ga]Ga-FAPI-46 uptake in a bleomycin-induced lung fibrosis murine model [55]. Lungs highlighted by yellow arrows.

**Figure 5 pharmaceuticals-17-00296-f005:**
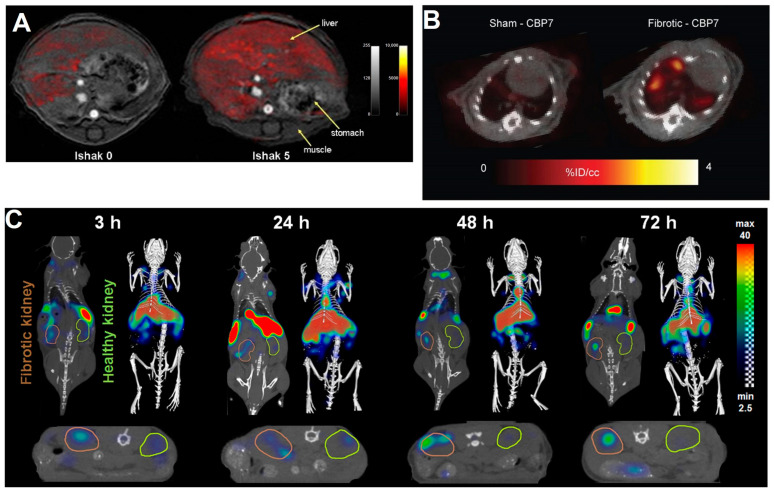
Different modalities for collagen imaging. (**A**) Imaging of EP-3533-enhanced MR in a CCl4-induced liver fibrosis mouse model [78]. (**B**) Imaging of [^64^Cu]Cu-CBP7 PET/CT in a bleomycin-induced lung fibrosis mouse model [86]. (**C**) Imaging of can35-Cy7 hybrid computed tomography-fluorescence molecular tomography (CT-FMT) in a renal fibrosis murine model [94]. The fibrotic kidney is encircled in brown (left) and the healthy kidney in green (right) in the coronal view.

## Data Availability

The original contributions presented in this study are included in the article. Further inquiries can be directed to the corresponding author.

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
