# Peer review of "Molecular Imaging of Fibrosis in Benign Diseases: An Overview of the State of the Art"

_pharmaceuticals, 2024, doi:10.3390/ph17030296_

Round 1
Reviewer 1 Report
Comments and Suggestions for Authors
The manuscript presents an intriguing and well-crafted exploration of molecular imaging, offering readers comprehensive insights into the latest advancements in this field. The authors adeptly integrate information on previous research, particularly focusing on molecular images. The manuscript is well-suited for publication in the journal.
Here are some of minor suggestions for the revision of the paper:
1. Clearly define what molecular imaging is and its historical background, identify gaps, and highlight the novelty of the current review.
2. On line 27, the description lacks precision; fibrosis itself may not lead to death.
3. From lines 28 to 31, please acknowledge that fibrosis can also manifest in the eyes and skin and should be emphasized in the review.
4. On line 62, about the inclusion of CT as a molecular imaging technique. It's a topic of debate, as Moritz F. Kircher et al. argue that that " Although CT offers high patient throughput and high-resolution imaging, molecular information is difficult to extract from CT images, as the images are based solely on differences in x-ray attenuation. (Radiology. 2012 Jun; 263(3): 633–643.)
5. Please consider whether spatial molecular imaging can be incorporated into molecular imaging technology. Authors may refer to the published paper: Nat Biotechnol 40, 1794–1806 (2022).
6. In the Molecular Mechanisms of Fibrosis section, please provide a brief overview of fibrosis-inducing factors, including growth factors, inflammatory factors, transcription factors, ECM, integrins, and epigenetic and genetic factors. Highlight TGFB as a major inducer.
7. Please address EMT as a early event of fibrosis. Acknowledge its importance and consider discussing whether EMT can be detected using molecular imaging techniques.
Reviewer 2 Report
Comments and Suggestions for Authors
The paper is of potential interest, but needs work to be appealing to non-specialists
1. The background section on fibrosis is obvious to people in the field. This can be shortened substantially. What is lacking is a background section on imaging and what is meant by probes. Readers are likely to be experts in fibrosis, and not imaging
2. The authors superficially describe the probes used to assess targets, without explaining WHY the specific probes are used to selectively image the targets.
3. Please explain HOW novel probes might be developed to identify probes of interest
4 a summary table of probes/targets is provided that is extremely extensive, but only a small number of probed/targets are explored in depth. I would rather that the authors focus, for example, on individual pathways and how probes might be developed to identify components of the pathway eg different parts of the integrin/TGFbeta/mechanotransduction pathway
Comments on the Quality of English LanguageThe English is readable, but non-standard. Professional editing is suggested
Reviewer 3 Report
Comments and Suggestions for Authors
The article titled "Molecular Imaging of Fibrosis in Benign Diseases: An Overview for the State-of-the-Art Progress" provides a comprehensive review of the current state and advancements in molecular imaging techniques for evaluating fibrosis in benign diseases. The authors have presented a well-structured and informative overview of the molecular pathways underlying fibrosis, potential targets, molecular probes, and challenges in the field. Overall, the article is well-written and provides valuable insights into the application of molecular imaging in fibrosis research.
The introduction provides a clear background on fibrosis and its significance in various diseases. However, it would be helpful to include a brief statement on the importance of early detection and repeatable monitoring in the management of fibrosis.
In Figure 1, the illustration of the application of molecular imaging in benign fibrosis diseases is informative. However, it would be beneficial to provide a brief description or caption explaining the different components of the figure for better clarity.
The section on molecular mechanisms of fibrosis is well-detailed and provides a comprehensive understanding of the processes involved. It would be valuable to include examples of specific diseases associated with fibrosis to further illustrate the relevance of the discussed mechanisms.
The section discussing molecular imaging techniques and their advantages is well-written. However, it would be advantageous to provide a brief summary or comparison of the different imaging modalities mentioned (ultrasound, MRI, CT, nuclear medicine techniques, and optical imaging) in terms of their strengths and limitations for fibrosis imaging.
The review of molecular probes developed for fibrosis evaluation is thorough and informative. It would be helpful to include some examples of specific molecular probes and their applications in preclinical or clinical studies to illustrate the progress made in this field.
Additional Questions:
What are the current challenges in translating molecular imaging techniques for fibrosis evaluation into clinical practice? Are there any specific hurdles that need to be overcome?
Could you provide some insights into the potential of molecular imaging in facilitating personalized treatment approaches for fibrotic diseases? How can molecular imaging contribute to patient stratification and the development of targeted therapies?
Are there any emerging molecular imaging techniques or probes that show promise in overcoming the limitations of current imaging modalities for fibrosis evaluation? What are the advantages of these new approaches?
In your opinion, what are the most critical areas of research and development needed to advance the field of molecular imaging in fibrosis evaluation? Are there any specific directions or collaborations that could accelerate progress in this field?
The article titled "Molecular Imaging of Fibrosis in Benign Diseases: An Overview for the State-of-the-Art Progress" provides a comprehensive and well-organized review of the current state and advancements in molecular imaging techniques for fibrosis evaluation. The authors have presented a clear overview of the molecular mechanisms underlying fibrosis, discussed potential targets, and provided insights into the development of molecular probes for fibrosis imaging. The article highlights the advantages of molecular imaging in visualizing biochemical processes and patterns of target localization at the molecular and cellular level, which can significantly contribute to the detection, characterization, and quantification of fibrosis.
The manuscript is well-written, and the information presented is relevant and valuable to researchers and clinicians working in the field of fibrosis. The inclusion of figures and illustrations enhances the understanding of the concepts discussed. However, there are a few minor areas that could be further improved, as mentioned in the specific comments above.
Based on the thorough review of the manuscript, I recommend its acceptance for publication with minor revisions.
Round 2
Reviewer 2 Report
Comments and Suggestions for Authors
Revisions are reasonable.